# Different Influences on "Wave Turbopause" Exerted by 6.5 DWs and Gravity Waves

Wei Ge [1], Zheng Sheng [2,3,*], Yingying Huang [4], Yang He [2], Qixiang Liao [2] and Shujie Chang [5]

[1] Unit No. 95806 of Chinese People's Liberation Army, Beijing 100076, China
[2] College of Meteorology and Oceanography, National University of Defense Technology, Changsha 410073, China
[3] Collaborative Innovation Center on Forecast and Evaluation of Meteorological Disasters, Nanjing University of Information Science and Technology, Nanjing 210094, China
[4] Beijing Institute of Applied Meteorology, Beijing 100029, China
[5] College of Ocean and Meteorology, Guangdong Ocean University, Zhanjiang 524088, China
* Correspondence: 19994035@sina.com

**Abstract:** "Wave turbopause" is defined as the mesospheric altitude level where the temperature fluctuation field indicates a substantial increase in wave amplitude in the vertical direction. It is similar to turbopause in seasonal and latitudinal variations, providing an almost global analysis of turbopause on the basis of satellite measurements rather than localized detection. Previous studies of "wave turbopause" were based on standard deviation of temperature, which is an integrated measure of wave activity. In this study, we distinguish different atmospheric waves and investigate their influences on "wave turbopause". By comparing the altitude of "wave turbopause" with peak height of amplitude (PHA) for gravity waves and 6.5 days waves (6.5 DWs), whose period is approximately 6.5 days in the mesosphere and lower thermosphere, we find that the seasonal variation in altitude of "wave turbopause" is higher at the winter pole and lower at the summer pole, correlated with PHA for 6.5 DWs but anti-correlated with PHA for gravity waves. We infer that gravity waves reach saturation and break at lower altitudes in the winter when Brunt–Vaisala frequency is also lower between 80 and 100 km altitudes. Finally, the results may imply that seasonal variations of the "wave turbopause" are driven mainly by 6.5 DWs.

**Keywords:** wave turbopause; gravity waves; 6.5 DWs

## 1. Introduction

From the ground surface to approximately 86 km in the U.S. Standard Atmosphere, turbulent mixing is much stronger than molecular diffusion and the separation effect of gravity. The constituents in the atmosphere can maintain their proportions due to turbulent mixing. Above 90 km to approximately 110 km, there is a transition layer from complete mixing to diffusion equilibrium. Turbulent mixing, molecular diffusion, photolysis of molecular oxygen, and ionization of the gas molecules coexist, and the latter three play leading roles above 120 km. The transition layer from turbulent mixing to diffusion equilibrium, or more precisely, the layer at which turbulent mixing coefficients are equal to the molecular diffusion coefficients, is called the turbopause. Several studies have also attempted to identify the turbopause as the altitude where molecular dissipation becomes stronger than turbulent dissipation [1] or where mixing ratios of constituents start changing [2].

Previous studies detected the turbopause by photographing trails released from rockets [3,4] or a mass spectrometer onboard rocket [5]. Medium frequency radar at a specific site has also been used [6]. In these studies, the altitudes of turbopause are rather different and vary from 80 to 120 km [7,8]. The differences in altitude of the turbopause, from 80 to 120 km, may occur for many reasons, such as different measurement techniques and

localized detections, or may be due to temporal and spatial variations in the turbopause itself. Distinguishing the cause of the differences need climatology of the turbopause. However, wide-range detections are not affordable and are hard to perform. A more practicable approach is needed.

In 2006, Offermann et al. [9] proposed a new turbopause conception called "wave turbopause". It is defined as the mesospheric altitude level where the temperature fluctuation filed indicates a substantial increase in wave amplitudes in the vertical direction. The method to identify "wave turbopause" is to calculate the intersection altitude of straight lines fitting the standard deviation of temperature detected by satellite. Thus, an almost global result is accessible. By comparing the result to a number of rocket measurements of the turbopause, the similarity between them shows the reliability of "wave turbopause" result to some extent [7]. However, the standard deviation is a "bulk parameter" or integrated measure of wave activity. It does not differentiate between the different types of waves, as also stated by Offermann et al. [9]. Specific contributions of various types of atmospheric waves are still worth studying.

Atmospheric waves include gravity waves, planetary waves, and so on. Gravity waves are seen as one of the major sources of dynamical variability in the atmosphere. Fritts et al. [10] addressed the link between instability and large-amplitude gravity waves in their review. He et al. studied the interaction between turbulence and gravity waves using a round-trip intelligent sound system and high-resolution GPS radiosonde soundings [11,12]. The interaction between gravity waves and turbulence is important, especially for atmospheric circulation such as quasi-biennial oscillation [13]. Gravity waves in the Martian thermosphere also exist and influence mean flow and turbulence, causing the diffuse transport of energy and momentum [14,15]. Planetary waves, also called Rossby waves, are main large-scale perturbations in the atmosphere, playing important roles in the transfer of energy and also in atmospheric circulation. Offermann et al. [16] compared the relative intensities of middle atmosphere waves, showing that gravity waves and planetary waves may be the most dominant contributors to the "wave turbopause". Further research on the influences of different atmospheric waves, especially gravity waves and planetary waves, may help us understand variations in the "wave turbopause" better.

In this study, we calculate the peak height of amplitude (PHA) for gravity waves and planetary waves and compare them with "wave turbopause" altitude to study their influences. Section 2 describes the SABER data and the methods we used to obtain residual temperature profiles due to waves and their PHA. A previous method to obtain the "wave turbopause" is also presented. Section 3 presents seasonal variations in the "wave turbopause" and the PHA for gravity waves and 6.5 DWs. Comparison between them shows both similarities and differences. A possible explanation for the differences is described in Section 4. Finally, conclusions are drawn in Section 5.

## 2. Materials and Methods

The SABER instrument was launched onboard the Thermosphere Ionosphere Mesosphere Energetics and Dynamics (TIMED) satellite in December 2001. One special strength of the SABER instrument is its temperature measurement, ranging from the tropopause to above 100 km. The 15 μm SABER infrared emission channel is used for this study, and Remsberg et al. [17] have given the details of the retrieval methods. The TIMED satellite performs yaw maneuvers approximately every 60 days, causing the SABER viewing geometry to shift between a northward (50°S to 82°N) and southward (82°S and 50°N) viewing geometry. SABER 2.0 data from 2004 to 2013 are used in this paper. In our analysis, the data are linearly interpolated to a fixed set of latitudes and altitudes. A latitude step of 1° is chosen from 50°S to 50°N, considering the continuity of data, and the altitude step is 1 km from 15 to 115 km.

### 2.1. Determination of the "Wave Turbopause"

Previously, the altitude of the "wave turbopause" was identified by two straight lines fitting the temperature standard deviation profile in different altitude regions. Each temperature standard deviation profile appears to consist of two parts: one with a moderate positive gradient at the lower altitudes (40–75 km) and another one with a steep gradient at the higher altitudes (90–110 km). The intersection kink of the two lines fitted by these two altitude regions is denoted as the "wave turbopause" [7,9]. John and Kishore Kumar [18] have also proposed that the standard deviation in a latitudinal band may have two different slopes instead of a single slope above the 95 km region in the mesosphere and lower thermosphere (MLT). Thus, they conducted two straight-line fits in the higher region, one at 90–100 km and another at 105–115 km, obtaining a "wave turbopause" layer. We also present the "wave turbopause" layer at the 50°S latitude band during January 2004 following this method as an example, which is shown in Figure 1.

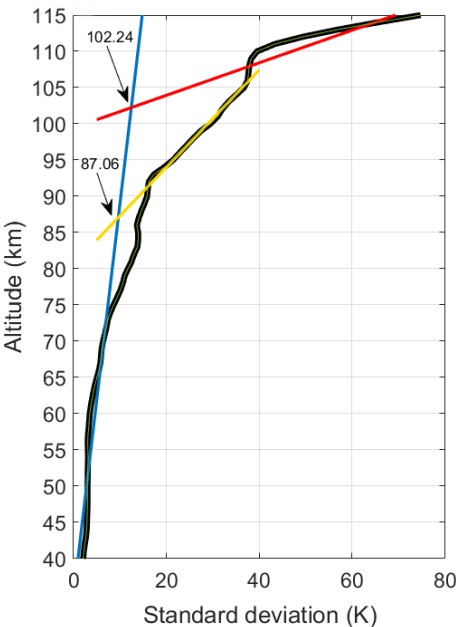

**Figure 1.** Height profile of standard deviation (black) from zonal mean temperature corresponding to 50°S latitude during January 2004. The blue, yellow, and red lines correspond to the straight-line fits at 40–75 km, 95–100 km, and 105–115 km, respectively. The straight lines intersect at the "wave turbopause" layer's lower and higher boundaries.

### 2.2. Identifying Temperature due to 6.5 DWs and Gravity Waves

The local background temperature profile includes the mean temperature and the temperature attributed to planetary gravity waves and tides, etc.

Firstly, we average the zonal temperature profiles separately for ascending and descending nodes, subtracting the local background temperature from the satellite observation to eliminate mean temperature and atmospheric tides. Ascending nodes and descending nodes represent the point at which an orbit crosses the ecliptic plane going north and south, respectively. Measurements at the same latitudes are taken at the same local time for ascending and descending nodes, causing a "phase-locking" of the tides, which means that all components of the migrating tides (diurnal and semidiurnal) alias to the daily zonal mean of the ascending and descending nodes, respectively [19].

Secondly, we obtain residual temperature profiles due to atmospheric waves by a least squares method [20,21]. A least square fitting to the term corresponding to frequency, σ, and wave number, s, is used to cover the short-period global-scale waves in the mesosphere.

$$y_i = A\cos[2\pi(\sigma t_i + s\lambda_i)] + B\sin[2\pi(\sigma t_i + s\lambda_i)] \tag{1}$$

The space-time series, $y_i$, are measured at UT $t_i$ (days) and longitude $\lambda_i$ (normalized by 360°). The wave amplitude is defined as:

$$R(\sigma, s) = \sqrt{A^2(\sigma, s) + B^2(\sigma, s)} \tag{2}$$

where $A^2(\sigma)$, $B^2(\sigma)$ represents the least squares estimates of $A$, $B$ at the frequency $\sigma$ and wave number $s$. The spectral range we choose here is similar to that chosen by Ern et al. [20]: zonal wavenumbers $-6$ to 6 and wave frequencies up to 1 cycle/day to include kelvin waves in the equatorial region. The time window is 31 days, with a 15-day step overlapped. Thus, for most altitude profiles, two estimates for the global-scale atmospheric background temperature are calculated and combined to minimize the boundary effects at the edges of the time windows. The time windows are non-overlapping when SABER changes from a northward to a southward viewing geometry or vice versa, with every viewing geometry lasting approximately 60 days. Spectral amplitudes are assumed to be due to global-scale atmospheric waves if the squared amplitude exceeds the white-noise level by a factor of at least 5. The reason for choosing this factor is also described by Ern et al. [20].

We obtain the planetary wave through the least-squares harmonic fitting above. The fitting result includes different planetary wave modes. The frequency ranges up to 1 cycle/day and the wavenumber is from $-6$ to 6, standing for westward or eastward propagation. In these modes, planetary waves (PWs) with a period of 5–7 days, traveling westward with zonal wave number 1, are known as 5-day waves (5 DWs). Meridional structures similar to 5 DWs have also been found in the mesosphere and the lower thermosphere, usually with a longer period, close to 6.5 days, and they are conventionally accepted as 6.5-day waves (6.5 DWs). Thus, the mode of 6.5 DWs is (6.5, W1). We extract the wave components with a frequency around 0.15 cycle/day and wavenumber $-1$ as the 6.5 DWs from the fitting results of global-scale atmospheric waves.

Thirdly, we obtained gravity waves by subtracting the fitting results of global-scale atmospheric waves from the temperature profiles obtained in the first step. Then, we average the residual temperature in two consecutive time windows because of the 15-day overlapping time windows. In addition, one correction of low-pass filtering with a vertical wavelength of 5 km is applied above 60 km to remove short-scale oscillations around the mesopause [20]. Figure 2 shows one example of the raw temperature profile and gravity wave perturbation after the above steps.

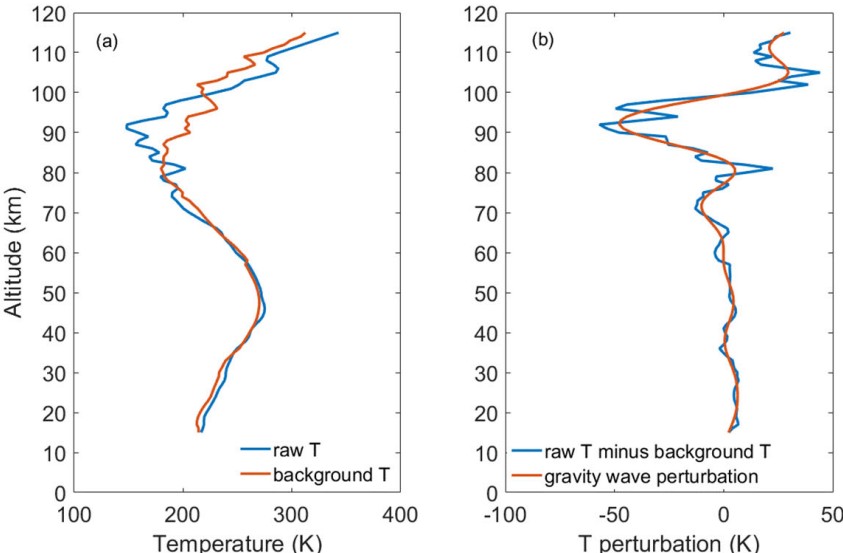

**Figure 2.** (**a**) Raw and background temperature profiles of one sounding sampled at (–50°, 337°) on 14 January 2004. (**b**) Difference between raw temperature and background temperature and the corresponding gravity wave perturbation profile (low-pass filtered).

Lastly, we use S-transform to analyze the gravity wave parameters. S-transform was introduced by Stockwell. This method replaces the wavelet basis function with a moving and scalable Gaussian window. The width of the Gaussian windows varies with the frequency [22]. We chose the largest amplitude and its corresponding wavelength as the parameters of the dominant gravity wave. The altitude region of the maximum amplitude is observed specifically for the dominant gravity wave. In addition, the longer vertical wavelength resulting from the leakage of tides and planetary waves still exists after subtracting the local background temperature, and we filtered out the oscillation with a vertical wavelength beyond 40 km for further cleanup. Figure 3 shows the gravity wave spectrum result corresponding to the sounding above.

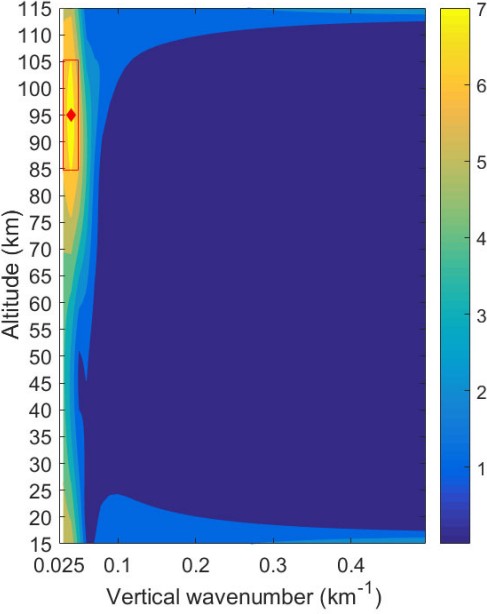

**Figure 3.** Amplitude–wavenumber spectrum from 15 to 115 km height (The vertical oscillation with vertical wavelength > 40 km is removed).

In Figure 3, the peak height of amplitude appears at approximately 95 km (red diamond), and the red rectangle region is the sustained increasing region. The upper and lower boundaries of the "wave turbopause layer" are approximately 102 km and 87 km, respectively, as determined by the intersections in Figure 1. The lower boundary of the "wave turbopause layer" is 8 km lower than the S-transform result. Atmospheric waves with different phase speeds will dissipate at different altitudes. Among these waves, gravity waves have relatively higher phase speeds and may propagate to higher heights [18].

## 3. Results

### 3.1. Seasonal Variation in the "Wave Turbopause" Layer

Firstly, we present the Month–Latitude variation in the "wave turbopause" layer obtained by the line fit based on temperature standard deviation in a latitude band, as shown in Figure 4.

Generally, the variations in higher and lower boundaries are similar and closely match those in the former studies. The maxima in the Southern Hemisphere appear from June to August regardless of the higher or lower boundary, while the counterparts in the Northern Hemisphere appear from December to February. In a word, the "wave turbopause" layer is higher at the winter pole and lower at the summer pole.

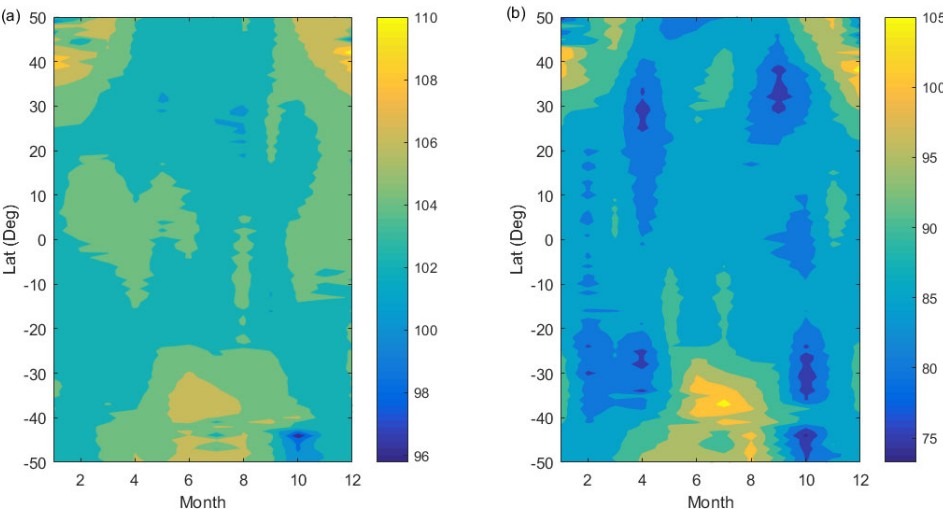

**Figure 4.** Month–Latitude Variations in the (**a**) higher boundary and (**b**) lower boundary of the "wave turbopause" layer by line fit averaged from 2004 to 2013.

### 3.2. Seasonal Variations in the Peak Height of Amplitude for 6.5 DWs

Amplitude attributed to 6.5 DWs is extracted from the global-scale atmospheric waves above. Here we present its Month–Latitude distribution in Figure 5.

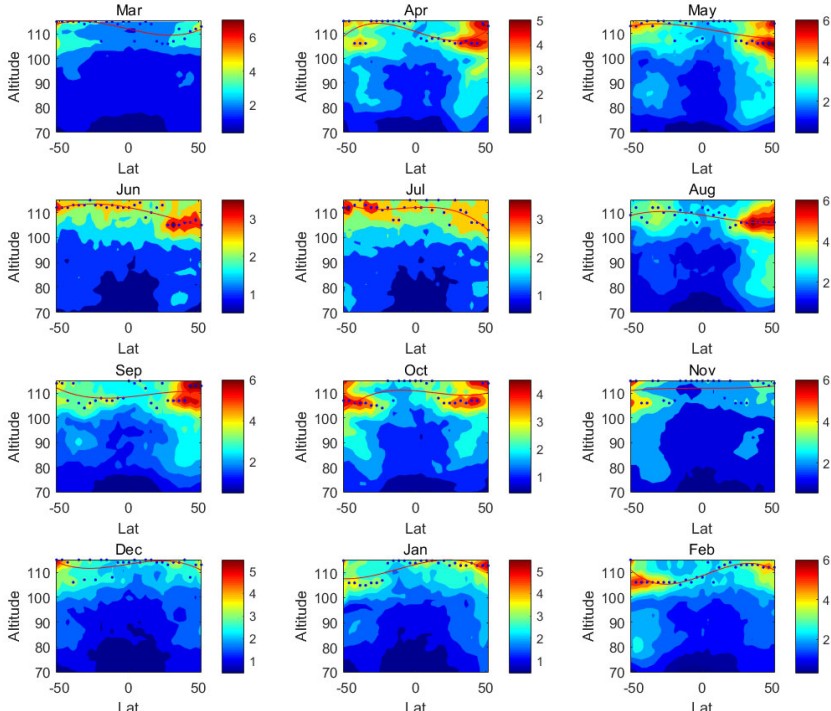

**Figure 5.** Monthly mean amplitudes of 6.5 DWs (red curved lines are third-order polynomial fitting of the peak height of amplitude at each latitude).

The red curved lines in Figure 5 are the third-order polynomial fittings of the peak height of amplitude at each latitude. The transition in the monthly mean results in December, January, February (DJF) and June, July, August (JJA) is obvious. In DJF, the peak height for 6.5 DWs in the Southern Hemisphere is lower than its counterpart in the Northern Hemisphere, while the result is reversed in JJA. This variation coincides with the seasonal variation in the "wave turbopause" layer.

### 3.3. Seasonal Variations in the Peak Height of Amplitude for Gravity Waves

In this section, we take the result in 2004 as an example, firstly in Figure 6.

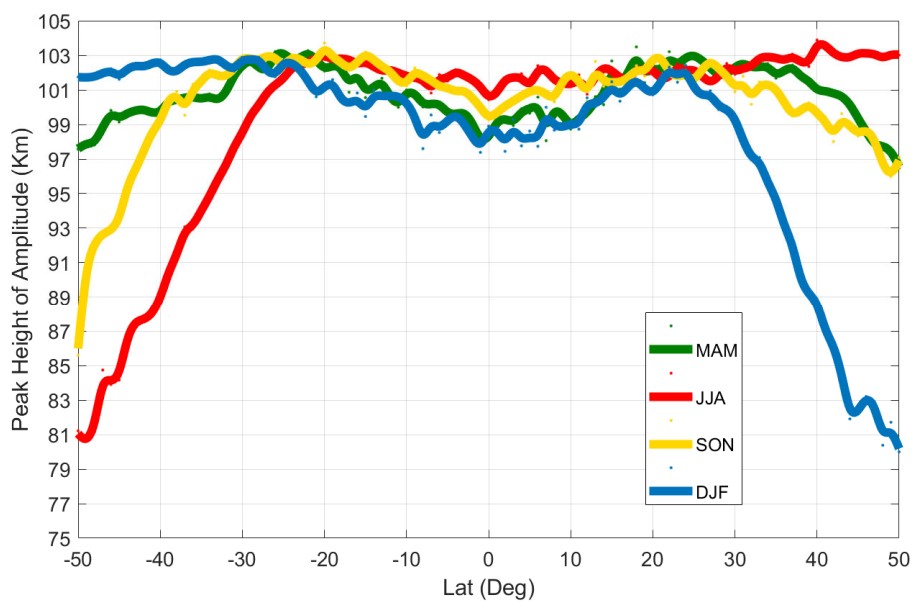

**Figure 6.** Latitudinal and seasonal variations in the peak height of amplitude for gravity waves in 2004.

Different from the seasonal variation in the "wave turbopause" layer and PHA for 6.5 DWs, the PHA for gravity waves is lower in the Southern Hemisphere from June to August and higher from December to February. In other words, the PHA for gravity waves is higher at the summer pole and lower at the winter pole. The same trend is seen for gravity waves from 2004 to 2013, and the results are shown in Figure 7.

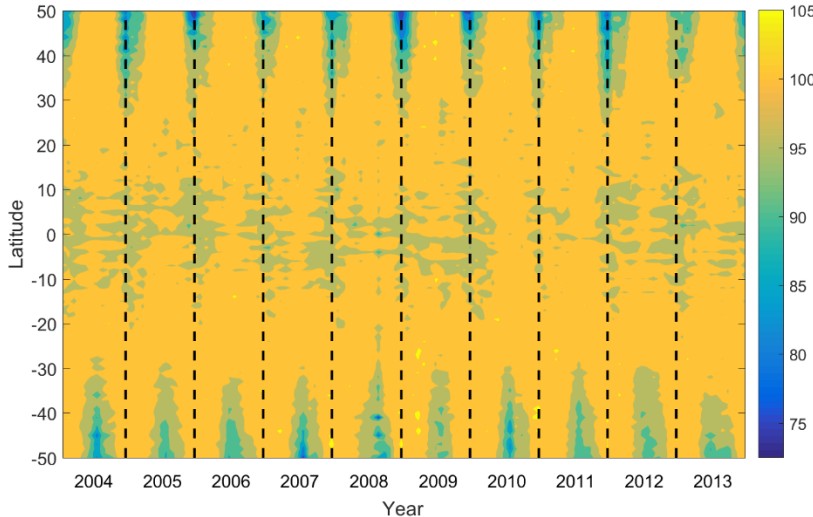

**Figure 7.** Distribution of the PHA for gravity waves along with latitude from 2004 to 2013.

## 4. Discussion

We infer that in the anti-correlation between PHA for gravity waves and the "wave turbopause" layer, PHA for 6.5 DWs may attribute to mesospheric stability variation, which may influence the saturation and break of gravity waves. Brunt–Vaisala frequency, as a proxy of atmospheric stability, has already been studied widely in previous research. Here we present the Brunt–Vaisala frequency distribution from previous studies, as shown in Figure 8.

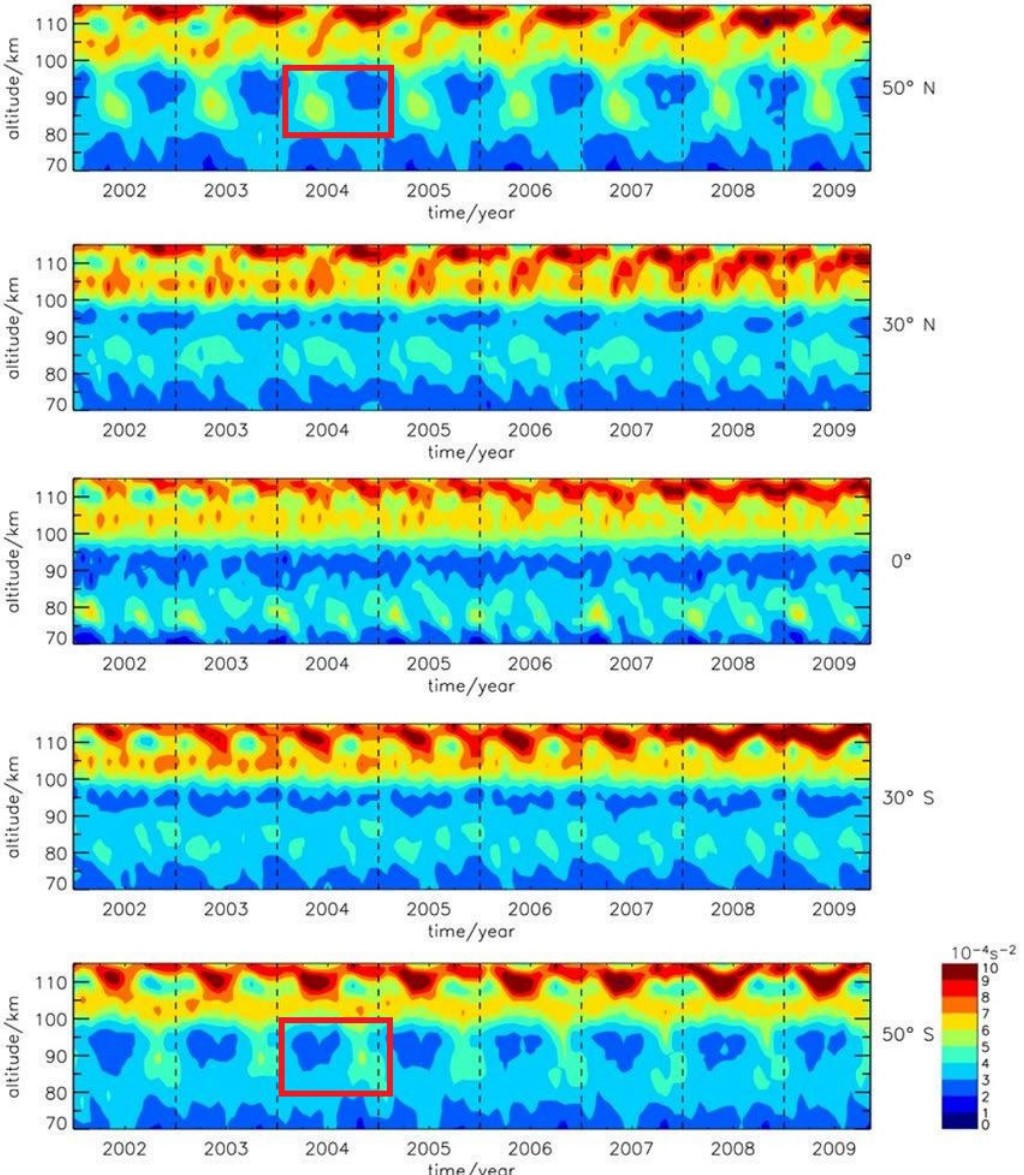

**Figure 8.** Distribution of the Brunt–Vaisala frequency along with altitude and latitude in 2002–2009 (see also Figure 3.7 in [23]).

The red rectangles obviously show the relative strength transition of Brunt–Vaisala frequency in the summer and winter. The frequency is higher in the summer pole and lower in the winter pole from 80 to 100 km. Higher frequencies imply a more stable atmosphere, able to sustain a continuous increase in gravity wave amplitude; the amplitude of gravity waves will increase exponentially when propagating upward until it exceeds the threshold and finally breaks. When the atmosphere is more stable, it can sustain a continuous increase in gravity wave amplitude, so that gravity waves may propagate to higher altitudes, where they reach the peak height of amplitude. This may explain why PHA for gravity waves is higher in the summer pole and lower in the winter pole.

## 5. Conclusions

This paper calculated the "wave turbopause" layer and peak height of amplitude for 6.5 DWs and gravity waves, respectively. Seasonal variations in these waves are also presented, and we find the following:



(1) Seasonal variations of the "wave turbopause" and the peak height of amplitude for 6.5 DWs are roughly the same. Both reach their maximum at the winter pole and their minimum at the summer pole.

(2) Seasonal variations of the "wave turbopause" and the peak height of amplitude for gravity waves are anti-correlated. PHA for gravity waves is lower in the winter pole and higher in the summer pole.

(3) A possible reason for the anti-correlation may lie on the Brunt–Vaisala frequency from 80 to 100 km. Relatively unstable atmosphere causes gravity waves to break at a lower height instead of propagating upward.

This paper only investigates 6.5 DWs as a reference of planetary waves; influences from other planetary wave modes need more investigation. Wave turbopause is seen as a climatology of the traditional turbopause according to former research. The findings regarding different influences in wave turbopause exerted by different waves may also help us study the relative roles of different waves in traditional turbopause regions.

**Author Contributions:** Conceptualization, W.G. and Z.S.; methodology, W.G.; validation, Y.H. (Yang He), Q.L., and S.C.; formal analysis, W.G. and Y.H. (Yingying Huang); investigation, W.G.; resources, Z.S.; data curation, W.G., Y.H. (Yingying Huang), and Y.H. (Yang He); writing—original draft preparation, W.G.; writing—review and editing, Z.S.; visualization, W.G. and Y.H. (Yingying Huang); supervision, Z.S.; project administration, Z.S.; funding acquisition, Z.S. All authors have read and agreed to the published version of the manuscript.

**Funding:** This research was funded by the National Natural Science Foundation of China, grant number 42275060.

**Data Availability Statement:** The SABER data are provided by http://saber.gats-inc.com/.

**Acknowledgments:** Thanks to Wenxiang Cao for providing the figure of the Brunt–Vaisala frequency variation, and the anonymous reviewers are gratefully acknowledged.

**Conflicts of Interest:** The authors declare no conflict of interest.

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
