# Peer review of "Different Influences on “Wave Turbopause” Exerted by 6.5 DWs and Gravity Waves"

_remotesensing, doi:10.3390/rs15030800_

Round 1
Reviewer 1 Report
The paper compares measures of the turbopause based on satellite measurements of wave activity. Methodology and results presented were sound and interesting, although there were a number of places where clarity could be improved.
L23: The meaning of 6.5 DWs should be introduced before the abbreviation is used.
L27: "And suppose that..." should be rephrased as a conclusion rather than an assumption.
L50: "wide-range" should replace "wild-range"
L67: round-trip intelligent sounding system should not be capitalised, consistent with previous usage.
L71-72: It's not clear why "Offermann et al." is mentioned in this sentence.
L116-117: Some explanation of ascending and descending nodes is needed.
Eq1: This equation does not show the relevant Fourier series, but only a single term corresponding to a wave number and frequency. Frequency \sigma and wave number s should be defined here also.
Eq2: Either the square root sign or exponent of R should be dropped. A, B, and R should be functions of lambda and sigma.
L132: This should read "wavenumbers -6 to 6", currently it's a bit ambiguous.
L148: The meaning of "(6.5,W1)" is not clear, please elaborate.
L180: It would be helpful to explain why higher phase speeds propagate to greater altitudes.
LL190: It may be better to state that variations in the lower boundary are "similar" rather than "consistent" (which might only mean they don't contradict one another).
L202: The meaning of terms like "DJF" and "JJA" needs to be provided.
L214-216: It's unclear what's being claimed in the last sentence of the paragraph, is it that the same trend is seen for gravity waves in years 2004-2013? Please revise.
L232: Further explanation should be provided regarding why increased stability increases the height where gravity waves break.
Reviewer 2 Report
see attachment

Reviewer 3 Report
Revision of "Different influences on “wave turbopause” exerted by 6.5DWs and 2 gravity waves" by Sheng et al. The manuscript presents an investigation of the activity of 6.5D wave and gravity waves in the temperature in the turbopause region using temperature measurements from the SABER/TIMED satellite. The topic is interesting and within the scope of "Remote Sensing". The manuscript is very concise, but the conclusions sound relevant scientifically. I have two moderate concerns that might be clarified before the acceptance for publication.Moderate points:
Lines: 105-107: The authors mentioned that they used the method by John & Kishore Kumar "at 50°S latitude band during January 2004", But it was not clear whether the method was applied for the whole observed period. May the authors explain it better?
Lines 154-157: the authors have performed a kind of filtering process to exclude oscillations with periods shorter than 5 km. Is there any physical reason to exclude this spectrum of gravity waves from the analysis?. In the MLT, the activity of gravity waves within this spectrum is very intensive as well and those waves can act in several processes in this region. Maybe, if this spectrum of gravity waves were taken into account, wouldn't the presented results be slightly different?
Minor points:
Line 66: et -> er
Line 70: waves -> wavs
Please, specify what is "s" in Equation 1.
--
Igo
